# Petrogenesis of Syn-Collisional Adakitic Granitoids and Their Copper Mineralization Potential in the North Qilian Suture Zone

Yuxiao Chen [1,2], Tianqi Zhang [1], Ying Cui [2,*] and Shuguang Song [2]

[1] Key Laboratory for Water Quality and Conservation of the Pearl River Delta, Ministry of Education, School of Environmental Science and Engineering, Guangzhou University, Guangzhou 510006, China; chyxiao@gzhu.edu.cn (Y.C.)

[2] MOE Key Laboratory of Orogenic Belts and Crustal Evolution, School of Earth and Space Sciences, Peking University, Beijing 100871, China; sgsong@pku.edu.cn

* Correspondence: cy0430@pku.edu.cn

**Abstract:** The petrogenesis of late Ordovician–early Silurian adakitic plutons in the North Qilian suture zone (NQSZ) and their copper mineralization potential remain poorly understood. Here we present a detailed study of the Heishishan (HSS) granodiorite–granite pluton, spatially associated with Cu–Au mineralization in the eastern section of the NQSZ. Zircon U–Pb dating confirms that the granodiorite–granite were formed at ca. 438–435 Ma, in association with a continental collision. Geochemically, the granitoids resemble low-Mg adakitic rocks featured by elevated Sr/Y and $(La/Yb)_N$ ratios with depleted MgO, Cr, and Ni concentrations, suggesting minimal mantle contribution. They are sodium rich with $K_2O/Na_2O < 1$, and have higher and more varied Sr/Y, but lower La/Yb than those from the continental lower crust. The $\varepsilon_{Hf}(t)$ values of zircon grains are positive and vary in a wide range of +2.0–12.7, indicating a heterogeneous source rather than a single arc basaltic source. They show moderately radiogenic Sr and Nd isotope compositions with initial $^{87}Sr/^{86}Sr$ ratios of 0.705101–0.706312 and $\varepsilon_{Nd}(t)$ values of +0.5–1.0, most likely a mixed source of the oceanic basaltic crust plus ca. 15–20% overlying sediments. The magmatic oxygen fugacity was relatively low as indicated by zircon Ce(IV)/Ce(III) ratios of 32–156, which is unfavorable for a large copper mineralization.

**Keywords:** adakite; porphyry Cu mineralization; zircon Hf isotope; syn-collisional granitoids; north qilian suture zone

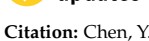



## 1. Introduction

The origin of adakite has received great attention for decades since it was proposed by Defant and Drummond [1]. Adakite was initially recognized as a type of arc lava and represents slab melts derived from young ($\leq 25$ Ma) and hot subducted oceanic crust [1], the origin of which then gained strong support from natural [2] and experimental studies [3–5]. It is defined by geochemical compositions (e.g., $SiO_2 \geq 56$ wt.%, $Al_2O_3 \geq 15$ wt.%, $Y \leq 18$ ppm, $Yb \leq 1.9$ ppm, $Sr \geq 400$ ppm, $Sr/Y > 40$, and $La/Yb > 20$) without detailed petrographic constraints. Therefore, igneous rocks with adakitic signatures, also termed "adakitic rocks", are found not only in modern subduction zones but also in ancient orogenic belts and cratons, and can also be generated by (1) the decompression melting of exhumed eclogites [6,7], (2) the melting/delamination of thickened lower continental crust (LCC) [8–13], (3) the fractional crystallization (FC) of normal basaltic/mafic andesitic magma with or without crustal assimilation [14–16], and (4) the magma mixing of mantle- and crust-derived melts [17].

Research on adakite/adakitic rocks has also been one of the hottest subjects in research on mineral deposits due to their strong connection to porphyry copper deposits [18–21].

In copper deposits where both adakites and non-adakitic rocks existed, mineralization is usually associated with adakites [18]. Several different explanations, i.e., the oxygen fugacity (oxidized), the source (initial enrichment of Cu in the oceanic crust) and composition (felsic and water rich) of the magma, and the compression tectonic environment, for the fertility of adakitic slab melts have been proposed, among which high oxygen fugacity and a derivation of the oceanic crust are thought to be two key factors [21–23].

The North Qilian suture zone (NQSZ) in northwest China is a typical Early Paleozoic suture zone that is composed of subduction-zone complexes including ophiolitic melanges, blueschists and eclogites, arc volcanic rocks, granitic intrusions, Silurian flysch formations, Devonian molasse, and post-Devonian sedimentary cover sequences [24–27] (Figure 1a). Recent studies have shown that adakitic granitoids in the NQSZ were mainly formed at ca. 457–430 Ma [16,28–31]. These adakitic rocks, e.g., the Heishishan and Quwushan plutons, are of great importance as they spatially show a close association with Cu–Au mineralization [32,33], and may offer essential clues to the prospecting of regional Cu–Au mineralization. However, their petrogenesis, particularly their derivations (i.e., the oceanic basaltic crust or the continental lower crust), is still controversial [16,28–30], which hinders our understanding of the potential of Cu–Au mineralization in the NQSZ.

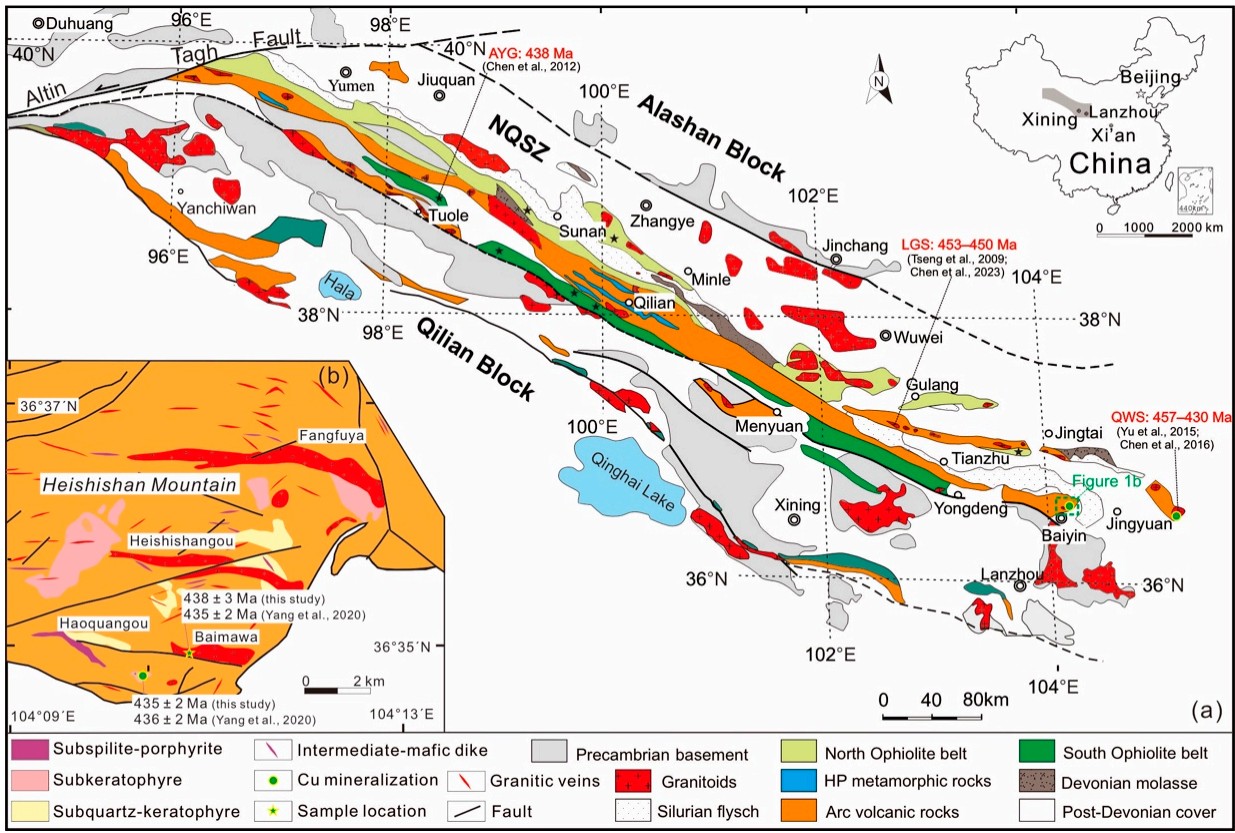

**Figure 1.** (**a**) Geological map of the North Qilian suture zone (NQSZ) showing major tectonic units and the Heishishan (HSS) pluton (modified after [30], the location of copper mineralization is after [33]), and (**b**) sketch map of the HSS granitic complex in the eastern NQSZ (modified after 1:50,000 geological map). Data sources include Chen et al., 2012 [6], Chen et al., 2016 [16], Tseng et al., 2009 [28], Yu et al., 2015 [29], Chen et al., 2023 [30], and Yang et al., 2020 [31].

In this study, we choose the Heishishan (HSS) pluton as it shows a close association in space with Haoquangou Cu (Au) mineralization [32]. We conduct U–Pb geochronology, trace element, and Lu–Hf isotope analyses of zircon grains, and bulk-rock geochemistry for the HSS adakitic granitoids, and discuss their petrogenesis and potential for Cu mineralization. We suggest that early Silurian adakitic granitoids in the NQSZ could be successfully

addressed by the partial melting of the oceanic basaltic slab, but the low-oxygen- fugacity of magmas is unfavorable for large copper mineralization.

## 2. Geological Setting and Petrography

The northwest-trending NQSZ is situated in the northern part of the Qinghai–Tibet Plateau, located between the Alashan block to the north and the Qilian block to the south and offset by the Altun Tagh fault to the west (Figure 1a). It has long been recognized as a typical Early Paleozoic suture zone that records a complete Wilson cycle from continental rifting to seafloor spreading, oceanic subduction, the extension of the back-arc basin, and to ultimately a continental collision and mountain collapse in the Neoproterozoic to early Paleozoic era [25,34,35]. It has been subdivided into three tectonic-magmatic subunits, i.e., the southern mid-ocean ridge (MOR)-type ophiolite belt (ca. 550–495 Ma), the middle arc magmatic belt (ca. 530–440 Ma), and the northern back-arc basin ophiolite–volcanic belt (ca. 517–450 Ma) (Figure 1a) [25,34–36].

Granitic plutons with zircon U–Pb ages of ca. 530–380 Ma are widespread in the NQSZ. They are characterized by a peraluminous granitic batholith and many medium-/high-K calc-alkaline I-type diorite–granodiorite–granite intrusions with minor tonalite–trondhjemite associations [37–42]. The Chaidanuo granitic batholith consists predominantly of peraluminous biotite monzogranite with biotite gneiss restites and mantle-derived enclaves, which was formed through the partial melting of Neoproterozoic granitic rocks with a minor contribution of mantle-derived magma during subduction initiation (~516–505 Ma) [40]. In addition, there are many adakitic plutons including the ca. 453–430 Ma Leigongshan tonalite [28,30], ca. 438 Ma Aoyougou trondhjemite [6], ca. 457–430 Ma Quwushan granodiorite [16,29], and ca. 436–435 Ma HSS granodiorite–trondhjemite [31], suggesting an extensive pulse of adakitic magmatism ca. 457–430 Ma in the NQSZ. Among them, the HSS and QWS adakitic plutons are spatially associated with porphyry Cu–Au deposits [32,33].

The HSS granitic complex is located ~10 km southwest of the Baiyin Cu–polymetallic ore field and comprises several intrusions (e.g., Haoquangou, Baimawa, Heishishangou, and Fangfuya) dominated by granodiorite and trondhjemite with a total area of ~10 km$^2$ [32] (Figure 1b). These intrusions occur as stocks, apophyses, and dykes and intrude into the Cambrian–Ordovician arc volcanic rocks that are bimodal with predominantly felsic units of 467–446 Ma [43,44] and minor mafic units of 465 Ma as indicated by zircon U–Pb ages [45,46]. The studied granodiorites were collected from the Haoquangou (HQG) and Baimawa (BMW) plutons (Figure 1b). The HQG pluton, located in ca. 3 km north of Baiyin City, is composed of medium-grained granodiorite and porphyritic trondhjemite that show a close relationship with Au–Cu mineralization based on element concentrations [32] (Figure 2a,b). Mafic microgranitoid enclaves (MMEs) are occasionally hosted by the granodiorite with transitional contacts (Figure 2b), and show a similar mineral assemblage of plagioclase, quartz, K-feldspar, and biotite, but a higher modal biotite with the granodiorite host. The BMW pluton lies ca. 1 km northeast of the HQG pluton and is dominated by medium-grained granodiorite (Figure 2c,d). The HQG and BMW granodiorites are composed of plagioclase (40–50%), quartz (35–25%), K-feldspar (5–10%), and biotite (<5%) (Figure 2e,f), and accessory minerals such as apatite, titanite, and zircon. Biotite is dark brown and partly replaced by muscovite or chlorite. Plagioclase is slightly sericitized with translucent cores surrounded by Na-rich, transparent rims. K-feldspar is mainly microline with gridiron twinning and occurs as anhedral grains between euhedral plagioclase crystals, indicating its crystallization in a late stage. Most muscovite is of secondary origin according to its occurrence (i.e., the replacement of the core of plagioclase and rim of biotite) and an overgrowth with epidote (Figure 2g,h). Overall, an overprint of greenschist facies metamorphism characterized by muscovite, chlorite, and epidote is evident in the HQG case, which is tightly associated with chalcopyrite (Figure 2g,h).

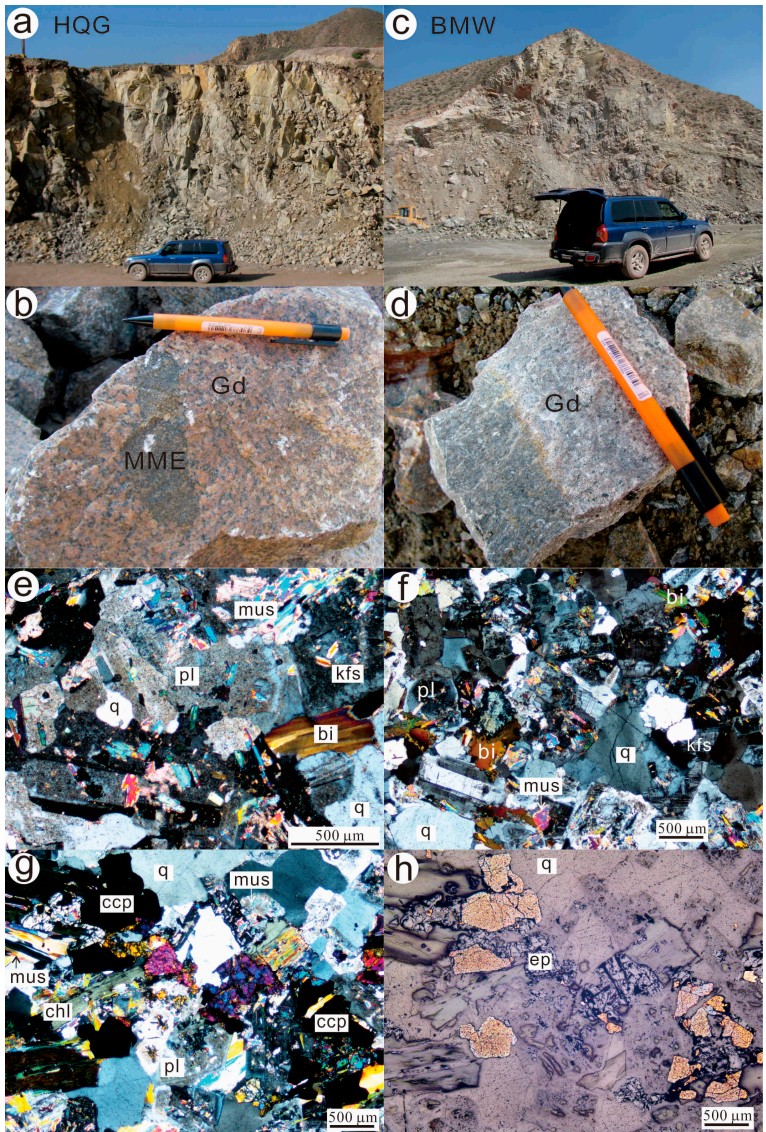

**Figure 2.** Field photos and microphotographs for the HQG (**a**,**b**,**e**,**g**,**h**) and BMW (**c**,**d**,**f**) granodiorites from the HSS granitic complex in the NQSZ. (**g**,**h**) Sample 12QL-183. Mineral abbreviations: biotite, bi; plagioclase, pl; K-feldspar, kfs; quartz, q; muscovite, mus; chlorite, chl; epidotite, ep; chalcopyrite, ccp. Please see the text for a description.

## 3. Analytical Methods

Zircons were separated from crushed rocks using conventional heavy liquid and magnetic techniques, then handpicked under a binocular microscope, mounted in an epoxy disc, and polished to half-sections. The internal zoning was examined using a cathodoluminescence (CL) spectrometer (Garton Mono CL3+) equipped on a Quanta 200F ESEM with a 2-min scanning time at conditions of 15 kV and 120 nA at the School of Physics, Peking University.

Measurements of U, Th, and Pb contents and isotopic ratios in zircons were conducted using an Agilient 7500ce ICP-MS equipped with a 193 nm laser housed at the Ministry of Education (MOE) Key Laboratory of Orogenic Belts and Crustal Evolution, School of Earth and Space Sciences, Peking University. The beam size of the analytic laser spot is approximately $32 \times 32$ μm. The calibrations for elemental concentration were carried out using NIST 610 glass as an external standard and $^{29}$Si as an internal standard. The detailed analytical procedure of LA-ICP-MS, the method for the corrections of the U–Pb isotope fractionation effect and common lead, and the calculation of $^{207}$Pb/$^{206}$Pb and $^{206}$Pb/$^{238}$U

ratios are similar to those described by Xia et al. [47]. Individual analyses are presented with 1σ errors in the data tables and in concordia diagrams. The uncertainties in mean ages are quoted at the 95% level. The analytical data are shown in Table S1.

Fresh rock samples were crushed into powders of 200 mesh in an agate mill. The major elements were analyzed using Leeman Prodigy inductively coupled plasma-optical emission spectroscopy (ICP-OES) at the Chinese University of Geosciences, Beijing (CUGB). The accuracy of the major elements is generally better than 2%. Trace element analysis was performed on an Agilent-7500a inductively coupled plasma mass spectrometry (ICP-MS) at Peking University, Beijing. The analytical accuracy indicated by the relative difference (RE) between the measured and recommended values is better than 5% for most elements. The analytical procedures for the major and trace elements were described by Song et al. [48] and Zhang et al. [49], respectively. The data are given in Table S2.

The separation and purification of Sr and Nd were conducted using conventional two-column ion exchange procedures in the ultraclean laboratory of the MOE Key Laboratory of Orogenic Belts and Crustal Evolution, School of Earth and Space Sciences, Peking University. The detailed procedure is the same as that described in Chen et al. [41]. The analysis was then conducted using a Triton Thermal Ionization Mass Spectrometer at the Institute of Geology and Mineral Resources, Tianjin. The detailed procedure follows that of Jahn et al. [50]. During the course of this study, the mean $^{87}Sr/^{86}Sr$ ratios for NBS-987 and BCR-2 were $0.710238 \pm 0.000005$ (2σ, $n = 3$) and $0.705016 \pm 0.000003$ (2σ, $n = 2$), respectively; the mean $^{143}Nd/^{144}Nd$ ratios were $0.512118 \pm 0.000006$ (2σ, $n = 5$) for JNDI and $0.512637 \pm 0.000006$ (2σ, $n = 2$) for BCR-2. The analytical data are shown in Table S3.

The in situ zircon Hf isotopic analysis was conducted using a Neptune multi-collector ICP-MS equipped with a Newwave UP213 laser in the MLR Key Laboratory of Metallogeny and Mineral Assessment, Institute of Mineral Resources, Chinese Academy of Geological Sciences (Beijing). In total, 15 analyses on the reference standard of zircon GJ1 yielded a weighted mean $^{176}Hf/^{177}Hf$ ratio of $0.282007 \pm 0.000007$ (2σ), in good agreement with the recommended $^{176}Hf/^{177}Hf$ ratio of $0.282000 \pm 0.000013$ (2σ) using a solution analysis method by Morel et al. [51]. The detailed procedure of analysis is similar to those described by Chen et al. [41]. The results are given in Table S4.

## 4. Results

### 4.1. Zircon U–Pb Geochronology

The granodiorite samples from the HQG (12QL-181) and BMW (12QL-190) plutons were chosen for LA-ICP MS zircon U–Pb dating. The zircon grains from both samples are colorless and euhedral to subhedral crystals with 100–200 μm in length and have length to width ratios of 2:1–4:1. They exhibit perfect oscillatory zoning in CL images, interpreted as a magmatic origin (Figure 3). Occasionally, they have oval, magmatic cores. In total, 25 spots on the rims of the zircon grains were analyzed for each sample. One date of $517 \pm 6$ Ma for the sample 12QL-181 could be affected by the old core of the zircon grain as indicated by CL images and, thus, is geologically insignificant. The 24 data-points for the sample 12QL-181 display Th contents of 155–446 ppm, U contents of 411–829 ppm, and Th/U ratios of 0.38–0.74, and yield $^{206}Pb/^{238}U$ ages of 428–443 Ma with a weighted mean $^{206}Pb/^{238}U$ age of $435 \pm 2$ Ma (MSWD = 3.7) (Table S1, Figure 3a), the same as the literature data of $436 \pm 2$ Ma within errors [31], interpreted as the crystallization age of the HQG pluton. The zircon grains from the granodiorite sample (12QL-190) give Th contents of 206–463 ppm, U contents of 306–692 ppm, and corresponding Th/U ratios of 0.49–0.78. They yield a weighted mean $^{206}Pb/^{238}U$ age of $438 \pm 3$ Ma (MSWD = 7.0, $n = 25$), slightly older than the date of $430 \pm 2$ Ma [31], representing the emplacement age of the BMW pluton (Table S1, Figure 3b). Therefore, the HSS granitic complex is suggested to have been emplaced at ca. 438–435 Ma.

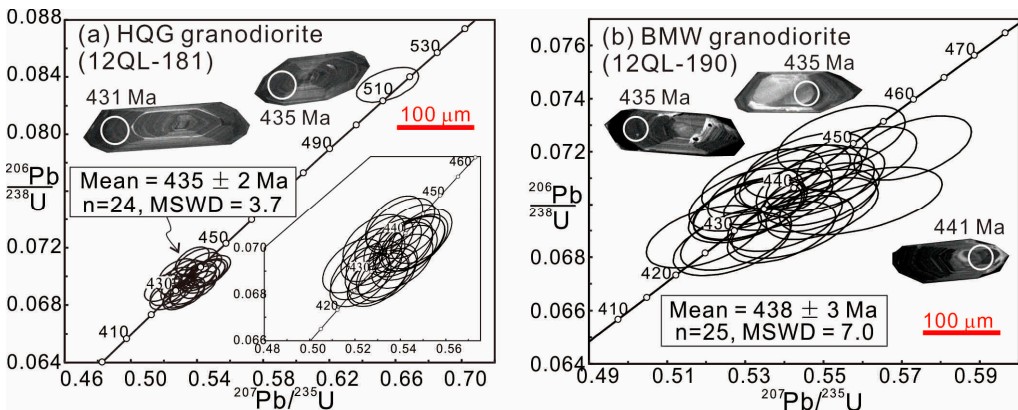

**Figure 3.** Representative CL images of zircon grains showing spots for LA-ICPMS U-Pb dating and concordia diagrams of zircon U-Pb age for the HQG (**a**) and BMW (**b**) granodiorite from the HSS pluton.

*4.2. Whole-Rock Major and Trace Elements*

Representative bulk-rock chemical compositions for the HQG and BMW granodiorites are listed in Table S2. The reference data are also shown for comparison [31,32]. Major element compositions are normalized to 100% on a volatile-free basis.

The HQG and BMW samples share great similarities in geochemical compositions (Table S2; Figures 4 and 5). They are characterized by high $SiO_2$ (68.9–72.3 wt.%), low MgO (0.96–1.36 wt.%), and a medium content of total alkali, plotting in the granodiorite field in the TAS diagram (Figure 4a). These granodiorites are sodic with high $Na_2O$ (3.43–4.06 wt.%), low $K_2O$ contents (1.48–3.01 wt.%), and $K_2O/Na_2O$ ratios (0.38–0.85), and belong to the middle-K calc–alkaline series (Figure 4b). They also have high $Al_2O_3$ contents (14.36–16.49 wt.%) and are weakly to strongly peraluminous with A/CNK values of 1.01–1.20 (not shown).

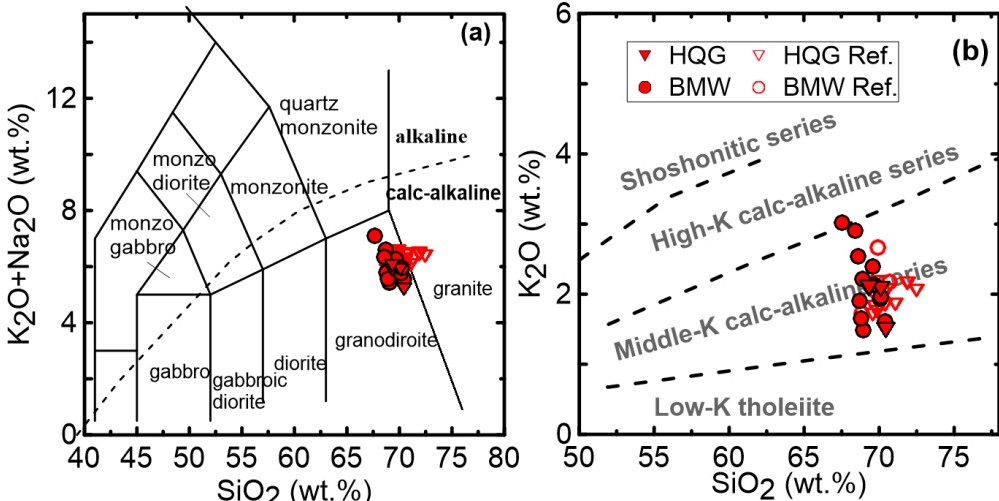

**Figure 4.** TAS diagram (**a**) and plot of $K_2O$ versus $SiO_2$ (**b**) for the HQG and BMW granitoids. Reference data for the HQG (shown as HQG Ref.) and BMW (shown as BMW Ref.) are from [31,32].

The HQG and BMW granodiorites also show similarly fractionated chondrite-normalized REE patterns, i.e., enriched in light rare earth elements (LREEs) but depleted in heavy REEs (HREEs) with $[La/Yb]_N$ of 14–41, with positive Eu anomalies (Eu/Eu* = 1.01–1.68) (Figure 5a). In the primitive mantle-normalized spider diagram, they are characterized by the enrichment of large ion lithophile elements (LILEs) with Rb, Ba, Sr, and Pb peaks, and the depletion of high field strength elements (HFSEs) with Nb, Ta, and Ti troughs

(Figure 5b). The granodiorites both exhibit adakitic signatures, e.g., high Sr (407–561 ppm), low Y (3.74–8.61 ppm) and HREE concentrations (e.g., Yb = 0.34–0.69 ppm), with high Sr/Y (56–119) and [La/Yb]$_N$ ratios (Figure 6a,b) [1].

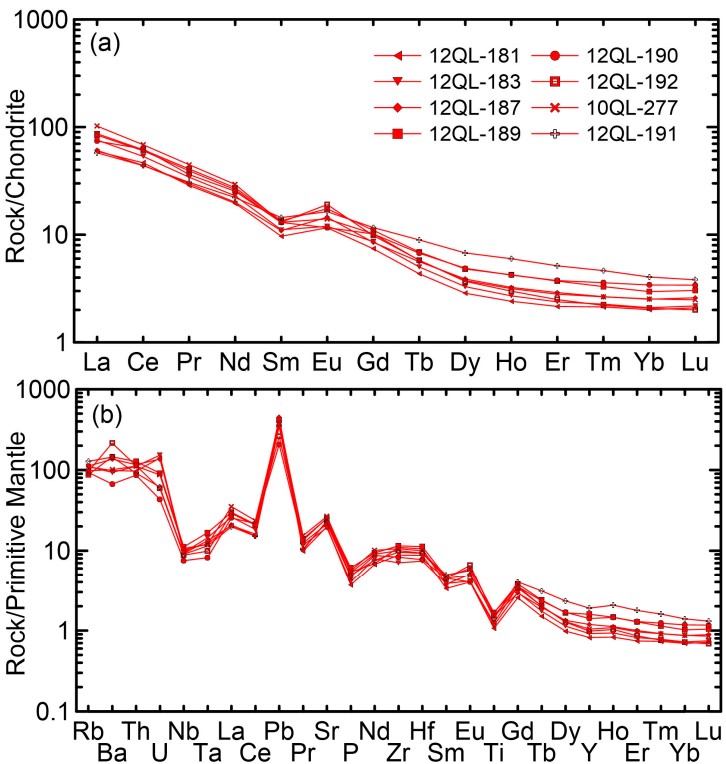

**Figure 5.** Chondrite-normalized REE patterns (**a**) and primitive mantle-normalized spider diagrams (**b**) for the HQG and BMW granodiorites. Normalization values are from [52].

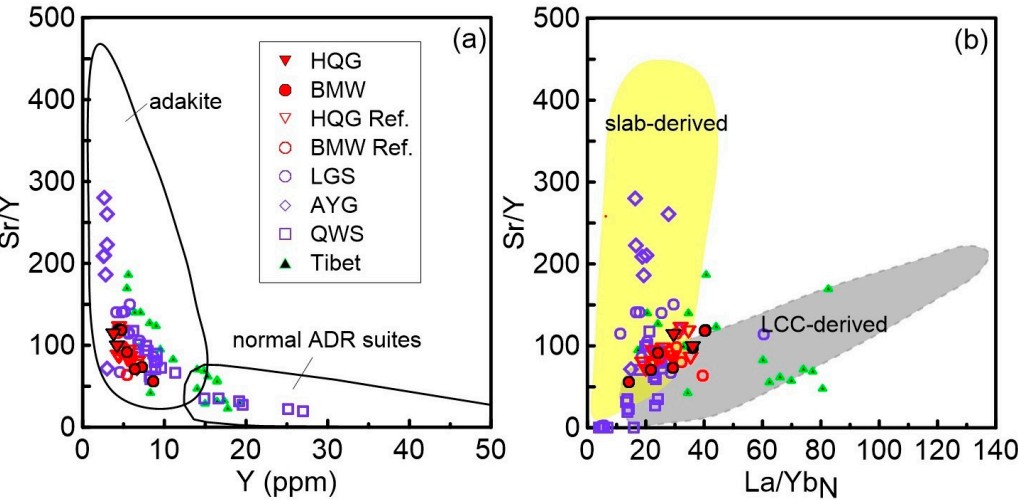

**Figure 6.** Diagrams of Sr/Y versus Y (**a**) and Sr/Y versus La/YbN (**b**) for the HSS adakitic granitoids. Panel (**a**) is after Defant and Drummond [1] and (**b**) after Liu et al. [53]. Data sources: Fields of slab-derived adakites include those in modern arcs from the GeoRoc database (https://doi.org/10.25625/2JETOA/FDAI5K), and LCC-derived adakitic rocks include those in the Tibet Plateau [10,13], Dabie Orogen [12] (Wang et al., 2007) and North China Craton (NCC) [9,17]. Adakitic plutons in the NQSZ including the LGS tonalite [28,30], AYG trondhjemite [6], QWS granodiorite–MME [16,29], and others, the same as in Figure 4.

### 4.3. Whole-Rock Sr and Nd Isotopes

The HQG granodiorites have initial $^{87}Sr/^{86}Sr$ ratios of 0.705177, and positive $\varepsilon_{Nd}(t)$ values of +0.8–+1.0, with two-stage depleted mantle Nd model ages [$T_{DM2}(Nd)$] of 1094–1105 Ma. Similarly, the BMW granodiorites have initial $^{87}Sr/^{86}Sr$ ratios of 0.705101–0.706312 and positive $\varepsilon_{Nd}(t)$ values of +0.5–+0.9, with two-stage depleted mantle Nd model ages [$T_{DM2}(Nd)$] of 1103–1135 Ma (Table S3; Figure 7). These data also resemble the literature data [31] (Figure 7).

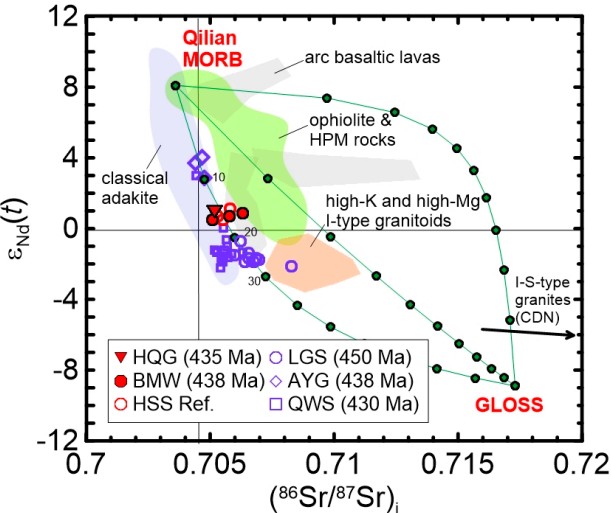

**Figure 7.** Initial Sr and Nd isotopes for the HQG and BMW granodiorites from the HSS pluton (modified after [30]). Data sources for adakitic plutons in the NQSZ are the same as those in Figure 6.

### 4.4. Zircon Hf Isotopes

Zircon Hf isotopic compositions for the HSS granitoids are calculated using their crystallization ages (Table S4; Figure 8). The zircon grains from the HQG and BMW granodiorites exhibit nearly identical but variable Hf isotopic compositions. In detail, the initial $^{176}Hf/^{177}Hf$ ratios, $\varepsilon_{Hf}(t)$ values, and [$T_{DM2}(Hf)$] are 0.282581–0.282860, +2.8–+12.7, and 609–1240 Ma for the HQG granodiorite, and 0.282555–0.282775, +2.0–+9.8, and 801–1297 Ma for the BMW granodiorite (Table S4; Figure 8).

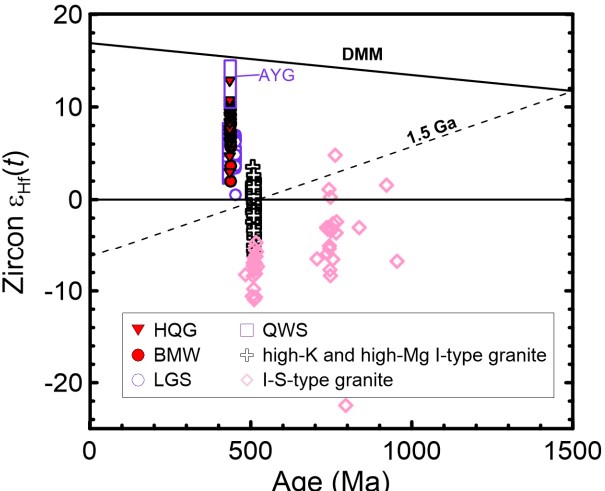

**Figure 8.** Zircon $\varepsilon Hf(t)$ values versus age (Ma) for the HSS adakitic granitoids. Data source: high-K and high-Mg I-type granite [41], I-S-type granite from the Chaidanuo granite [40]. The AYG field represents the slab-derived trondhjemite in the NQSZ (unpublished data). The others are the same as those in Figure 6.

## 5. Discussion

### 5.1. Adakitic Signature Obtained through Partial Melting

Intermediate to felsic rocks with adakitic signatures, e.g., high Sr/Y and La/Yb ratios, can be generated through or modified by (1) the fractional crystallization of basaltic-andesitic magmas [14–16], (2) the partial melting controlled by source compositions and melting conditions [1,9,12,54], or (3) mixing of mantle- and crust-derived melts [17].

The mantle contribution is suggested to be insignificant for the HSS adakitic granitoids based on the three lines of evidence below. First, field investigations show that MMEs are locally hosted by the HQG granodiorites with transitional contacts, and have indistinguishable mineral assemblages but with higher mode volumes of biotite than the hosts (Figure 2). This indicates that they are probably biotite aggregates or cumulates [16,40] rather than mixed products of crust- and mantle-derived magmas [17]. Second, the HSS adakitic granitoids including the HQG granodiorite and trondhjemite, and BMW granodiorite [31,32] show low MgO contents (0.96–1.36 wt%) and Mg numbers (48–51), resembling those of experimental melts from metabasalts at 1–4 GPa [4,5] and sediment melts at 2–4 GPa [55] (Figure 9a), precluding an important role of the mantle input through magma mixing [17] or mantle metasomatism during the magma ascent [56].

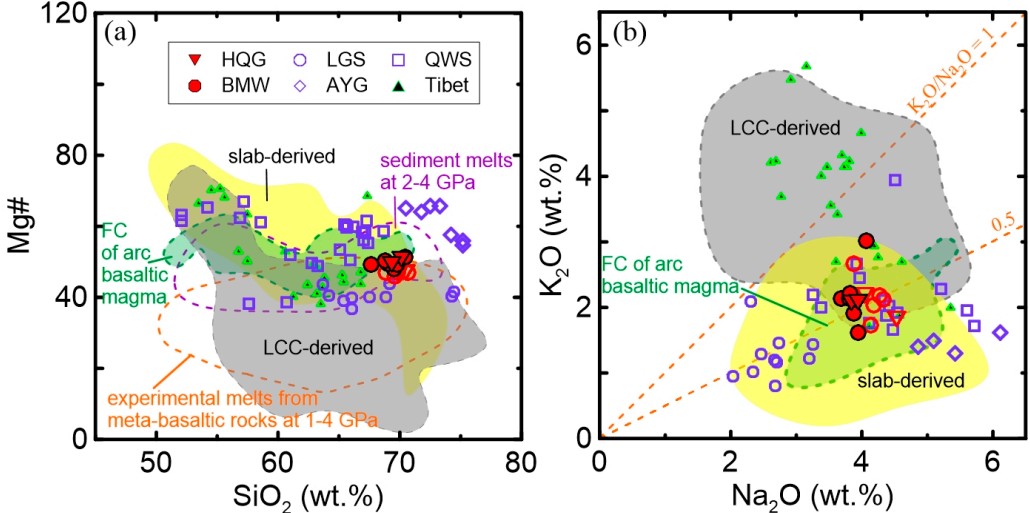

**Figure 9.** Plots of (**a**) Mg# versus SiO$_2$ and (**b**) K$_2$O versus Na$_2$O for the HSS adakitic granitoids. Data sources: experimental melts from sediments [55] and meta-basaltic rocks at 1–4 GPa [4,5]; fractional crystallization (FC) of arc basaltic magma from [14,15]; others are the same as in Figure 6. Symbols are the same as in Figures 4 and 6.

Third, the fractional crystallization model of mantle-derived basaltic–andesitic magmas involving an amphibole-/garnet-dominant mineral assemblage is also unlikely.

The FC model requires tectonically voluminous basic associations derived from the metasomatized mantle wedge in arc settings or abundant MMEs of cumulate origin hosted in plutons [14–16]. The HSS adakitic granitoids lack a complete compositional spectrum from normal arc basic rocks to high Sr/Y intermediate-felsic rocks (Figure 10a) and are characterized by high SiO$_2$ (67.52–70.37 wt%) and low to moderate K$_2$O (1.48–3.01 wt%) (Figure 4b), distinct from those generated through an FC of arc basaltic parental magmas that are generally low in SiO$_2$ and middle to high K calc–alkaline [14,15]. On the other hand, the role of garnet or amphibole in fractionation is insignificant as manifested by the absence of either mineral (Figure 2) and trends of element concentrations (e.g., La, Dy, and Yb) and ratios (e.g., Sr/Y, La/Yb, and Dy/Yb) with increasing silica content (Figure 10). Except for one sample, the HSS adakitic granodiorites show a trend of decreasing Sr/Y, La/Yb and Dy/Yb with increasing silica content, accompanied by an increase in Dy and Yb. This could reflect the fractionation of minor plagioclase but not amphibole or garnet as the removal

of amphibole or garnet would lower the Dy and Yb in the derivative melt. Plagioclase fractionation is also supported by the wide range of Eu* ratios and their positive correlation with the Sr concentration (Figure 10d). The Eu* ratio also decreases with decreasing P (not shown), indicating that apatite fractionation is unimportant in the HSS case (also see Figure 9d in [16]).

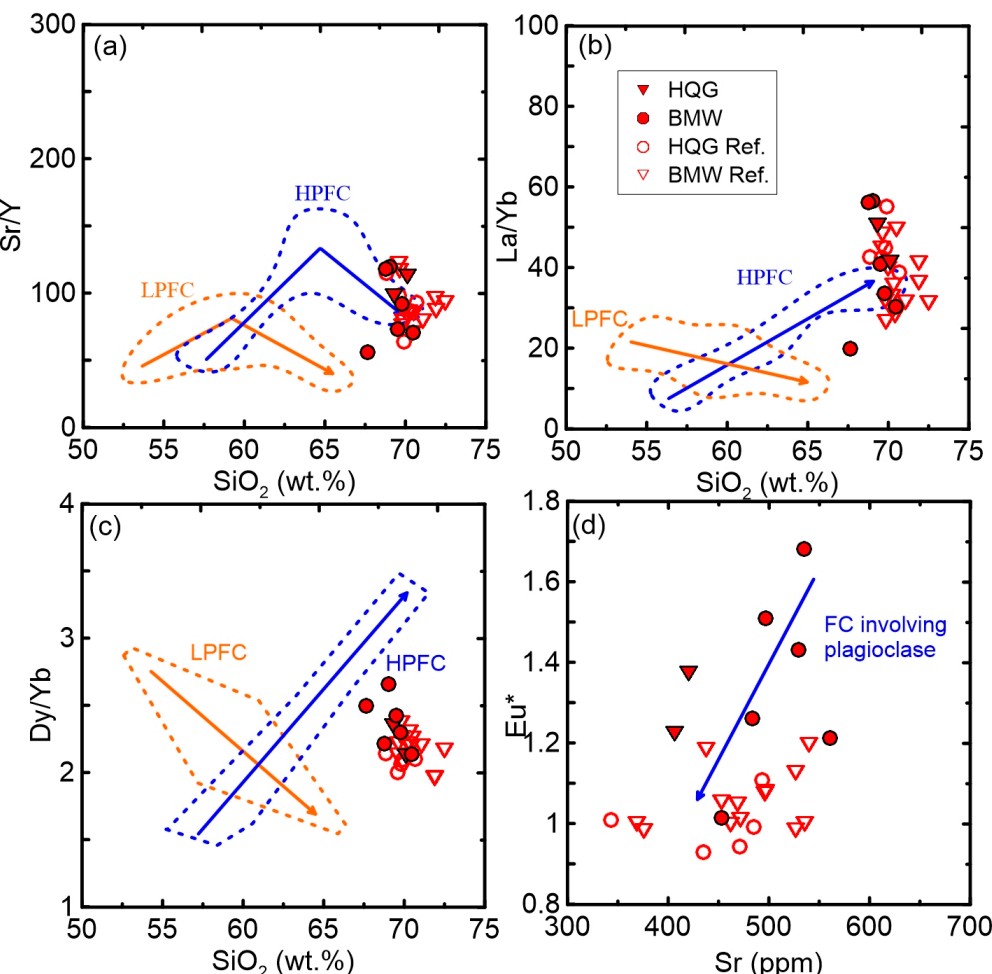

**Figure 10.** Diagrams of (**a**) Sr/Y versus SiO$_2$, (**b**) La/Yb versus SiO$_2$, (**c**) Dy/Yb versus SiO$_2$, and (**d**) Eu* versus Sr. Trends for low-pressure fractional crystallization (LPFC, orange field) and high-pressure fractional crystallization (HPFC, blue field) in (**a–c**) are after [14] and [15], respectively. Symbols are the same as in Figure 4.

To discriminate the partial melting/magma mixing processes from fractional crystallization, graphic models of highly (H) and moderately (M) incompatible elements (DH << 0.2–0.5, H = Rb, K, Th, and La; DM << 1, M = Nd and Sm) that have been considered as one of the most effective and robust tools were introduced [57,58] (Figure 11). The HSS adakitic granitoids generally have steep, straight lines in the plots of La/Sm versus La (CH/M vs. CH) and a straight line that does not pass through origin 0 in the plot of K$_2$O versus Rb (CH1 vs. CH2) (Figure 11). These incompatible element diagrams suggest that in terms of incompatible elements, the HSS adakitic granitoids are largely controlled by the partial melting/magma mixing process rather than crystal fractionation. Based on the observations mentioned above, the HSS adakitic granitoids were probably produced by the partial melting of crustal rocks (also see below for discussion).

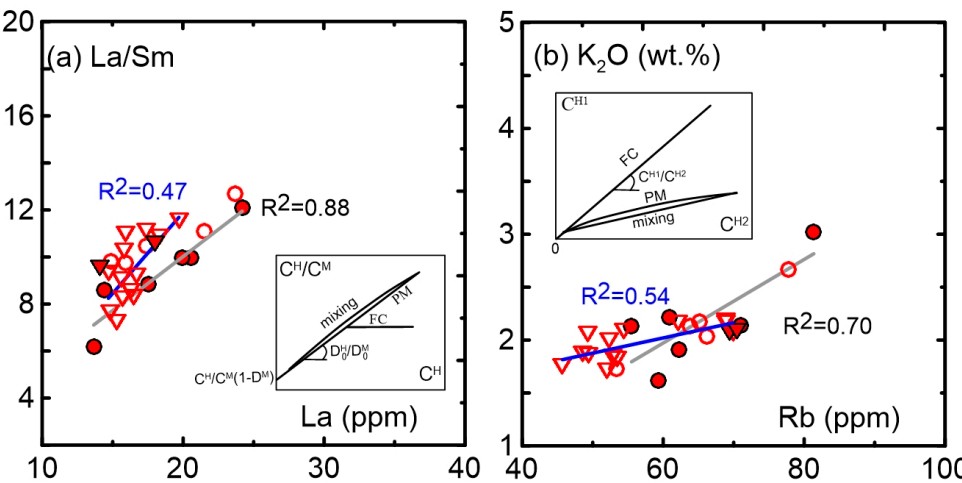

**Figure 11.** Plots of (**a**) La/Sm versus La and (**b**) $K_2O$ versus Rb for the HSS adakitic granitoids in the NQSZ. Symbols are the same as in Figure 4. Additionally shown are schematic diagrams (CH/M versus CH diagram in (**a**), with H and M being highly and moderately incompatible elements; CH1 versus CH2 diagram in (**b**), with H1 and H2 being two highly incompatible elements) with curves showing compositional trends of magmatic processes including partial melting (PM), fractional crystallization (FC), and mixing (MIX).

### 5.2. Partial Melting of the Oceanic Crust

For the partial melting model, the possible candidate includes basaltic protoliths from (1) the ancient lower continental crust (LCC), (2) the newly underplated lower crust (NLC), and (3) the subducted oceanic crust. Low-Mg adakitic granitoids (ca. 457–435 Ma), including the LGS tonalite, the studied HSS granodiorite–trondhjemite, and the QWS granodiorite, have been previously considered as partial melts of the thickened NLC [29,31,32,59], which then experienced thinning through delamination to generate the QWS MME-bearing high-Mg adakitic granodiorite (ca. 430 Ma) in a post-collisional setting [29,31]. Although this model seems plausible and applicable to the HSS adakitic granitoids, it has more difficulties than certainties.

Tectonically, the Paleo–Qilian ocean is suggested to have begun its subduction at ca. 530 Ma, and to have closed at ca. 445 Ma followed by the continental collision, as indicated by the ophiolites, eclogites, and arc basaltic–andesitic lava in the NQSZ [25,35]. Accordingly, the ca. 438–435 Ma HSS adakitic granitoids, together with the ca. 438 Ma AYG trondjemite and ca. 430 Ma QWS granodiorite [6,16], are best explained as a magmatic activity in response to continental collision rather than in a post-collisional setting [29,31]. In fact, a continuous lithospheric extension and orogenic collapse (e.g., delamination) in the NQSZ would not have occurred at >400 Ma, which was responsible for the generation of a series of diorite–granodiorite–granite plutons with ages of ca. 400–360 Ma [60,61].

Geochemically, the HSS adakitic granitoids display low $K_2O$ concentrations and $K_2O/Na_2O$ ratios (0.38–0.85) averaged at 0.55, obviously different from the old LCC-derived adakitic rocks defined by those in the North China Craton and Dabie orogen [9,12,17] (Figure 9b). The sodium-rich feature in adakitic granitoids generated in a continental arc setting has also been explained as a derivation from an NLC [11], which has led researchers to propose the partial melting model of thickened NLCs for the Neogene adakitic rocks in Tibet [10]. As shown in Figure 9b, adakitic rocks in Tibet contain both sodic and potassic types, and the latter also show high $K_2O$ with $K_2O/Na_2O$ ratios >1 that plot in the field of LCC-derived adakitic rocks [10,13]. This suggests a partial melting of the different components, including either an ancient or a newly underplated basaltic protolith, in a thickened continental crust setting. However, the HSS adakitic granitoids, together with other adakitic rocks (e.g., LGS, AYG, and QWS) in the NQSZ, roughly agree with the classic slab-derived adakites in the circum-Pacific subduction

zones (http://georoc.mpch-mainz.gwdg.de/georoc/), suggesting their derivation from an oceanic basaltic slab. Moreover, compared with LCC-derived adakitic rocks, the slab-derived adakites have systematically higher and more varied Sr/Y coupled with lower La/Yb in the Sr/Y versus [La/Yb]$_N$ diagram [53] (Figure 6b). This is mainly because (1) the altered oceanic crust usually has a high Sr content, (2) the average La/Yb of the lower continental crust (5.3) is more than six times higher than that of the average MORB (0.8), and (3) La is several times more mobile than Yb during plate subduction, such that the subducted slab has even lower La/Yb [62]. In this respect, adakitic granitoids in the NQSZ are characterized by a relatively low La/Yb but a variable and high Sr/Y, which are comparable to classic slab-derived adakites (Figure 6b). This further suggests their derivation from an oceanic basaltic slab rather than the thickened LCC.

In terms of Sr–Nd isotopes, the HSS adakitic granitoids partly overlap with arc basaltic lavas in the NQSZ, which has been considered as evidence for the thickened NLC model [31]. However, it should be noted that the zircon Hf isotopes of the HSS adakitic granitoids vary in a wide range with eHf(t) values of +2.0–+12.7 (Figure 8), suggesting their derivation from a mixed source rather than a single basaltic protolith ultimately from an enriched subarc mantle. In addition, comparisons of adakitic rocks, ophiolites, and arc basaltic rocks in the NQSZ using bulk-rock Sr–Nd and zircon Hf isotope data provide important insights into the protolith of adakitic rocks. The HSS and other adakitic granitoids in the NQSZ display isotopes that are significantly less radiogenic than those of arc high-K and high-Mg calc–alkaline I-type granitoids (Figures 7 and 8), the latter of which are mainly sourced from the sub-arc lithospheric mantle with continental materials [41]. In the Sr–Nd diagram, the adakitic rocks are obviously different from the enriched arc basaltic lavas (Cambrian–Ordovician) or the depleted basaltic protoliths from ophiolites, eclogites, and blueschists. Instead, they define a curved trend between the Qilian MORB and GLOSS end members, which can be explained as a mixture of basaltic oceanic crust and sediments (Figure 7). Convincingly, the ca. 450 Ma LGS adakitic tonalites are characterized by abundant inherited cores with old ages of 470–2733 Ma in zircon grains. The short time interval (<20 Ma) between the formation of adakitic toanlites and deposition of sediments (<470 Ma) suggests the contribution of subducted sediments to the melting source region at depths of >40 km for adakitic melts [30]. Therefore, the Sr–Nd–Hf isotope evidence points to a significant input of a depleted mantle with a sediment contribution rather than an NLC originating from an enriched sub-arc lithospheric mantle with/without continental materials. Calculations based on the two-component mixing model of Sr–Nd isotopes indicate ca. 5–10% and ca. 15–20% of sediment contribution to the AYG [6] and HSS, respectively.

### 5.3. Implication for Copper Mineralization in the NQSZ

Porphyry Cu–Au deposits around the world are mostly distributed in active convergent margins, e.g., the circum-Pacific, Paleo–Asian and Tethis–Himalaya metallogenic belts [63]. Previous studies have shown that high oxygen fugacity and slab melting are two key factors controlling the formation of porphyry Cu mineralization [20,21,64–66]. Slab melting is favorable for porphyry Cu mineralization because (1) the contents of Cu, Au, and S in the oceanic basaltic crust are much higher than those in the LCC and mantle, which are inherited by the derivative melt, and (2) the oxygen fugacity of the subduction zone is approximately two orders of magnitude higher than that of the mantle and LCC. High oxygen fugacity can greatly improve the solubility of sulfur in magma, which is conducive to the transformation from sulfide in the source to sulfate in the melt, thus greatly increasing the Cu content in the initial magma. On the other hand, as a moderately incompatible element, Cu in the melt can be further enhanced during magmatic evolution as sulfide remains unsaturated in an oxidized magma [21–23].

The HSS and QWS adakitic plutons have been invoked to be associated with porphyry Cu–Au mineralization [32,33]. In this study, the finding of chalcopyrite in the HQG granitoids that are overprinted by a greenschist facies metamorphism also support a genetic

link between porphyry Cu (Au) mineralization and adakitic rocks in the NQSZ. However, the metallogenic potential of the adakitic rocks is still unclear. As mentioned above, recent studies show that the adakitic rocks in the NQSZ more likely represent partial melts of the oceanic basaltic slab [6,16,30], which is favorable for porphyry Cu mineralization [32,33].

To further assess the metallogenic potential of these adakitic plutons, we estimate the oxygen fugacity of the magma using the zircon Ce(IV)/Ce(III) ratio following the method of [64]. The zircon grains from the HQG and BMW granodiorites have Ce(IV)/Ce(III) ratios of 64–156 and 32–128, respectively (Table S1). These Ce(IV)/Ce(III) ratios of zircons are similar to those in the A-type granitic plutons (2–198) that are related to the W–Sn mineralization in the Nanling Range [67], but significantly lower than those in the Dabaoshan Cu-bearing porphyry (300–800) [23,68], suggesting a low magmatic oxygen fugacity. The reduced nature of the HSS adakitic melt may be related to a small amount of sediment input (ca. 15–20%) into the melting source as manifested by Sr–Nd isotope modeling. As the form of sulfur in the melt is controlled by the oxygen fugacity, copper precipitates prematurely from the magmatic system with the crystallization of sulfide under low-oxygen-fugacity conditions, which hinders the further enrichment of copper during magmatic evolution and ultimately the formation of large-scale mineralization.

## 6. Conclusions

(1) The HSS adakitic pluton was formed ca. 438–435 Ma in response to a continental collision.

(2) Bulk-rock geochemistry, Sr–Nd isotopes, and zircon Hf isotopes indicate its derivation from the oceanic basaltic crust with minor contribution of subducted sediments (ca. 15–20%).

(3) The zircon Ce(IV)/Ce(III) ratio indicates a low oxygen fugacity for the adakitic melt, which is unfavorable for large copper mineralization.

**Supplementary Materials:** The following supporting information can be downloaded at: https://www.mdpi.com/article/10.3390/min13070892/s1, Table S1: LA-ICP MS zircon U–Pb data for the HSS adakitic granitoids; Table S2: Major and trace element data for the HSS adakitic granitoids; Table S3: Whole-rock Sr–Nd isotopic composition for the HSS adakitic granitoids in the NQSZ; Table S4: Zircon Lu–Hf isotopes for the HSS adakitic granitoids.

**Author Contributions:** Conceptualization, methodology, and writing—original draft preparation, Y.C. (Yuxiao Chen); investigation and sample collection, Y.C. (Yuxiao Chen) and T.Z.; formal analysis, T.Z.; writing—review and editing, Y.C. (Yuxiao Chen) and Y.C. (Ying Cui); funding acquisition, S.S. and Y.C. (Ying Cui). All authors have read and agreed to the published version of the manuscript.

**Funding:** This research was funded by the Foundation of Guangzhou City for basic research projects (202201010468), the National Natural Science Foundation of China (Grant Nos. 41903022), and Earth Critical Zone and Eco-geochemistry (PT252022024).

**Data Availability Statement:** All data, models, or codes that support the findings of this study are available from the corresponding author upon reasonable request.

**Acknowledgments:** We would like to thank Xin Xu and Mengjue Wang for field sampling, Li Su for helping with the whole-rock geochemical analyses, Fang Ma for the LA-ICP MS dating, Kejun Hou and Chunli Guo for the MC-ICP MS in situ zircon Hf analyses, and Guozhan Li and Wenping Zhu for the Sr–Nd isotopic PT252022024 analyses.

**Conflicts of Interest:** The authors declare no conflict of interest.

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
