# Peer review of "Petrogenesis of Syn-Collisional Adakitic Granitoids and Their Copper Mineralization Potential in the North Qilian Suture Zone"

_minerals, doi:10.3390/min13070892_

Round 1

Reviewer 1 Report

This paper focuses on an early Silurian adakitic granodiorite-trondhjemite pluton in the Qilian Orogen. It contains zircon U–Pb age and Lu-Hf isotopic data and whole-rock elemental and Sr–Nd isotopic data. The main objectives of the paper are to constrain the petrogenesis of the Heishishan adakitic granitoids including the derivation and processes of magmatism and their copper mineralization potential. The authors suggest that chemical variations of the HSS granitoids are mainly controlled by the partial melting process and they are derived from a mixed source of oceanic basaltic crust plus ca. 15-20% overlying sediments during continental collision. However, the magmatic oxygen fugacity is relatively low, which is unfavorable for large copper mineralization. As an overall comment, I consider that this paper can be accepted for publication after minor revisions.

Main Issue:

1.      The title is incompleted and needs to be revised (Line 2):

Can it be “Petrogenesis of syn-collisional adakitic granitoids and their copper mineralization potential in the North Qilian suture zone”?

2.      The derivation of the HSS adakitic granitoids needs to be clarifed.

It is suggested to be from a heterogeneous source with significant input of the depleted mantle rather than a single arc basaltic source (line 20). However, the contribution of recycled sediments to the source has been manifested as well (line 379-390).  Please clarify the source nature.

Detailed comments:

Line 228: Figure 5b the text of x-axis is squished together and needs to be revised.

Other comments have been given in the text.

This is a well-written paper on the geochemistry of felsic rocks in the Qiilan Orogen. I am pleased with the writing style and the way in which the geochemical data have been presented.

Reviewer 2 Report

This study examines the Heishishan (HSS) granodiorite-granite pluton and its association with the Baiyin Cu-Au deposits in the North Qilian suture zone (NQSZ). The authors aim to shed light on the petrogenesis of late Ordovician adakitic plutons in the NQSZ and their potential for copper mineralization.

The authors employ detailed analyses, including zircon U-Pb dating and geochemical assessments. The granitoids exhibit sodium-rich characteristics and low-Mg adakitic rock features, with depleted mantle contributions. The zircon grains indicate a heterogeneous source with significant input from the depleted mantle and a mixed source of oceanic basaltic crust and overlying sediments. However, the relatively low magmatic oxygen fugacity observed suggests limited copper mineralization potential.

This paper provides valuable insights into syn-collisional adakitic granitoids and their relationship to copper mineralization in the North Qilian suture zone. The methodologies used, including zircon U-Pb dating and geochemical analyses, contribute to our understanding of the HSS granodiorite-granite pluton's petrogenesis. The findings regarding the source characteristics and the implications for copper mineralization offer important considerations for further exploration efforts. Researchers interested in adakitic granitoids and copper mineralization in suture zones will find this paper informative.

I am not an expert in adakitic magmatism, but this paper seems to present a comprehensive study of adakitic magmatism in China. I would like the conclusions to be better written, emphasizing the adakitic model that you favor and making them more accessible to a general audience. Additionally, it is necessary to reduce the use of acronyms and ensure that they are properly translated to improve readability. Another issue is that the references are not formatted correctly, lacking appropriate capitalization. Please review and rewrite them accordingly. Furthermore, I believe that a native speaker could enhance the text by improving its structure, language, and addressing the numerous typographical errors that I have identified.

Specific comments:

Abstract: Are the adakitic rocks derived from crustal contributions rather than mantle sources? Could you please provide a more thorough explanation of this in the text?

Section 2

Line 103-105: Could you provide the percentage of minerals for the petrographic description of the HQG pluton?

Section 4.2:

Please explain the difference between slab-derived and LCC-derived adakitic granitoids, as shown in Figure 6b, and provide more discussion on this topic in the text. Where are your rocks located?

Please consider that positive Eu anomalies can also be attributed to amphibole fractionation at the source. Are there any amphiboles present in your rocks? (see Kay, 1978, Kay, R. W. (1978). Aleutian magnesian andesites: melts from subducted Pacific Ocean crust. Journal of Volcanology and Geothermal Research, 4(1-2), 117-132.)

Section 4.3: Please expand and provide a better explanation for this section.

Figures

What does HQG Ref and BMW Ref stand for? (they appear in several figures, please check)

In Figure 5, which ones represent the HQG and the BMW?

I believe that a native speaker could enhance the text by improving its structure, language, and addressing the numerous typographical errors that I have identified.

Reviewer 3 Report

This paper is so wonderfully well written!

Author Response

All the suggestions listed in the original manuscript have been accepted. Thank you. Please see the new version of the manuscript.